# Nanoparticles as Drug Delivery Vehicles for People with Cystic Fibrosis

**DOI:** 10.3390/biomimetics9090574

**Published:** 2024-09-22

**Authors:** Eoin Hourihane, Katherine R. Hixon

**Affiliations:** 1Thayer School of Engineering, Dartmouth College, Hanover, NH 03755, USA; eoin.g.hourihane@dartmouth.edu; 2Geisel School of Medicine, Dartmouth College, Hanover, NH 03755, USA

**Keywords:** cystic fibrosis, drug delivery, nanoparticles, metals, polymers, lipids, treatments, antibiotics, FDA

## Abstract

Cystic Fibrosis (CF) is a life-shortening, genetic disease that affects approximately 145,000 people worldwide. CF causes a dehydrated mucus layer in the lungs, leading to damaging infection and inflammation that eventually result in death. Nanoparticles (NPs), drug delivery vehicles intended for inhalation, have become a recent source of interest for treating CF and CF-related conditions, and many formulations have been created thus far. This paper is intended to provide an overview of CF and the effect it has on the lungs, the barriers in using NP drug delivery vehicles for treatment, and three common material class choices for these NP formulations: metals, polymers, and lipids. The materials to be discussed include gold, silver, and iron oxide metallic NPs; polyethylene glycol, chitosan, poly lactic-co-glycolic acid, and alginate polymeric NPs; and lipid-based NPs. The novelty of this review comes from a less specific focus on nanoparticle examples, with the focus instead being on the general theory behind material function, why or how a material might be used, and how it may be preferable to other materials used in treating CF. Finally, this paper ends with a short discussion of the two FDA-approved NPs for treatment of CF-related conditions and a recommendation for the future usage of NPs in people with Cystic Fibrosis (pwCF).

## 1. Cystic Fibrosis Introduction

Cystic Fibrosis (CF) is characterized as a life-shortening, genetic disease, and the majority of people with Cystic Fibrosis (pwCF) die of lung infection and inflammation as a result of the disease [1]. As of 2024, approximately 40,000 individuals in the US, and another 105,000 outside of the US, are living with CF [2]. Historically, the median survival age for pwCF has been incredibly low, with children born in 1954 only expected to live to 4 to 5 years [3]. However, modern improvements in treatments and therapies have drastically improved and continue to improve survival rates, such that children born between 2019 and 2023 are currently expected to have a median survival age of 61 years [4].

CF is a genetic disease caused by a mutation in the Cystic Fibrosis Transmembrane Conductance Regulator (CFTR) protein, located on chromosome seven at position 7q31 [5]. CFTR is an important protein found throughout the body in exocrine glands and certain epithelial cells and serves as an ion channel to transport chloride (Cl^−^) and bicarbonate (HCO_3_^−^) from one side of the cellular membrane to the other [6]. This flux of ions from one side of the cellular membrane to the other forces sodium ions (Na^+^) to move via an electrochemical gradient and water to move through osmosis, both following chloride ions and bicarbonate, to maintain concentration and charge equilibria. The CFTR direction of net transport depends on the location in the body. For example, in sweat glands, CFTR transports chloride ions and bicarbonate from sweat (extracellularly) back into the cell (intracellularly), which causes water and sodium resorption into the cells, as sodium moves down the electrochemical gradient and water moves from an area of lower concentration of chloride, bicarbonate, and sodium to an area of higher concentration of chloride, bicarbonate, and sodium [7,8]. In the lungs, CFTR transports chloride ions and bicarbonate in the opposite direction, intracellularly to extracellularly, which causes water and sodium to move from inside to outside the cells. This movement of water and sodium from inside to outside the cells, mediated by CFTR, is one of the major mechanisms for hydrating the mucosal layer in the lungs [9]. A schematic of this process is shown in Figure 1.

In pwCF, the CFTR protein is mutated, resulting in either a reduced or complete lack of transport of chloride ions and bicarbonate. In the lungs, this causes the mucosal layer to dehydrate as compared to an excess of chloride ions and bicarbonate in sweat [9,11]. The increased amount of chloride ions in sweat is one of the major ways in which newborn children are tested to see if they have CF [11,12]. In fact, throughout human history, mothers were known to lick a newborn child’s forehead as part of cleaning ceremonies. If a newborn child tasted salty, it was assumed that the child would die soon, providing potentially the first historical anecdotes of CF [13].

Overall, six classes of mutations are recognized in pwCF [14] which can clarify what is happening at a cellular level; however, this is largely unhelpful in treating the majority of pwCF clinically [15]. This is due to the drastic variations in clinical presentations of CF depending on other genetic factors (e.g., ENaC, an important sodium transport channel found in lung epithelial cells) and because CF is a recessive mutation, where people may possess two different classes of mutations on their chromosomes. Furthermore, there are approximately 2000 recognized mutations in CFTR that cause CF [15], and different mutations can cause different severities. For example, a mutation resulting in a less functional CFTR likely has better clinical outcomes than a mutation causing no CFTR with respect to being transcribed. Many of the approximately 2000 variants are unclear, and the disease liability of only about 25 CFTR mutations are well known [16]. The six classes of CFTR mutations, however, are important in designing medications to treat the underlying mechanisms of CF, such as CFTR modulators. The six classes of CFTR mutations are as follows [14]:

(i)No CFTR protein is produced: This can be caused by mutations in the start codon for CFTR or mutations in promoter regions causing a lack of transcription.(ii)CFTR protein does not make it to the membrane: This occurs when certain mutations cause the CFTR protein to not fold correctly and thus be marked for degradation before it can make it to the membrane.(iii)CFTR is non-functional: Here, CFTR properly folds and is transported to the membrane but has a loss of function mutation in residues that are needed for ion transport.(iv)CFTR is less functional: While CFTR is mostly functional, it is not able to transport as many chloride ions and bicarbonate as required to sustain homeostasis.(v)Less CFTR is produced: CFTR is fully functional and transcribed but is not transcribed at high enough levels.(vi)CFTR is less stable: CFTR is fully functional and transcribed at the correct levels but is marked for degradation earlier than it should be, causing less CFTR to remain at the membrane.

The second class of CFTR mutations (in which CFTR is transcribed but does not make it to the membrane) is the most common cause of CF in the US due to the deletion of phenylalanine 508 (ΔF508) [14]. Specifically, it is estimated that ~70% of all alleles causing CF are characterized by this specific mutation [16,17]. This mutation causes CFTR to misfold and be marked for degradation, resulting in no CFTR protein being embedded in the membrane. CFTR modulators, which have been a breakthrough in the treatment of CF, function by rescuing this misfolded protein and allowing it to be transported to the cell membrane. Once there, the CFTR protein, while less functional, can transport ions effectively enough to alleviate many symptoms. CFTR modulators are effective against ΔF508 mutations, among other mutations that cause protein misfolding. However, patients who have a mutation that falls within one of the other five classes or who have protein misfolding that cannot be rescued are not effectively treated with CFTR modulators; therefore, these patients must continue seeking other treatment options [14]. In cases where CFTR is not transcribed or cannot be effectively rescued, gene therapy is the only effective treatment method, but no FDA-approved gene therapies for CF have been approved to date.

## 2. Pulmonary Mucosal Layer

As stated previously, the most harmful aspect of CF is the effect it has on the lungs and, specifically, the mucosal layer. To understand the effect CF has on the mucosal layer, it is first important to understand what the mucosal layer is, its function, and the biological pathology in pwCF.

The mucosal layer in the lungs is an extremely thin (on the order of microns) gel-like lining that coats the surface of airway epithelial cells. Beneath the mucosal layer is the periciliary layer that contains the cilia of certain airway epithelial cells surrounded mostly by water. The mucosal layer serves first and foremost as a physical barrier between any inhaled debris or bacteria and the airway epithelial cells. Mucus in the lungs is constantly moved from distal to proximal airways (i.e., bronchioles to bronchi to trachea) through the process of ciliary beating, in which the cilia move rapidly back and forth to transport the mucosal layer. After the mucus moves up the trachea, it then travels down the esophagus and into the stomach for digestion. This clearance of mucus from the airway and into the esophagus is a major mechanism by which the lungs clear the airway; specifically, this results in the physical transportation of unwanted pathogens, particulates, chemicals, and anything else that may be harmful from the lungs into the esophagus [9].

The mucosal layer is composed of airway epithelial cells which form a 50/50 mosaic of secretory (creating mucus) and ciliated (transporting mucus) cells. The mucosal layer is also formed by submucosal glands in larger airways (>2 mm). Overall, the mucosal layer is ~97% water in healthy individuals, where the final 3% is what gives mucus its important structural and functional properties. Specifically, this final 3% is composed of various molecules, including antimicrobial, immunomodulatory, and protective molecules; however, the most important solids excreted into the mucosal layer are mucins [9,18].

Mucins are large glycoproteins that serve to provide the mucosal layer with the required rheological properties [18]. These rheological properties of mucus are paramount, with ineffective flow resulting in unwanted particulates remaining within the lungs. The two most expressed mucins in airway epithelial cells are MUC5B and MUC5AC [9,18]. MUC5B is composed of 5762 amino acids and weighs approximately 596 kDa [19], while MUC5AC is 5654 amino acids long and weighs approximately 585 kDa [20]. For reference, the average length of a eukaryotic protein is 472 amino acids [21], demonstrating the sheer size of these proteins. Further, this exceedingly large size is further increased by the amount of O-linked glycosylation sites and their ability to polymerize. This glycosylation and polymerization of mucins is what grants the mucosal layer its gel-like structure and viscoelastic properties [22].

In pwCF, as stated previously, one of the most damaging effects of a dysfunctional CFTR protein is the resulting change in hydration of the mucosal layer. Specifically, in pwCF, the amount of water in the mucosal layer drops from ~95% to ~80% water content. This causes the periciliary layer to shrink and the cilia to collapse, hindering the transportation mechanism from the lungs into the esophagus, allowing particulates caught in the mucosal lung to remain. This buildup of mucus causes many problems in the lungs, such as infections, inflammation, mucus plugs, and collapsed alveoli [9]. A schematic highlighting these differences between a normal lung and CF lung is shown in Figure 2.

## 3. Nanoparticles

As stated previously, the cause of death for pwCF is most commonly lung infection/complications caused by CF [1,7]. Due to this, there has been a surge of interest in creating targeted vehicles for drug delivery to the lungs. While recent advances in CFTR modulators have lessened the importance of this due to their effectiveness, e.g., in preventing declining lung function [24], targeted drug delivery is still needed where infection remains the leading cause of death for pwCF, and CFTRs are not effective in every patient [1,14]. A pulmonary route for drug delivery is ideal for targeting the lungs in pwCF, as it requires a lower dosage and there are less off-target effects vs. oral or intravenous administration [25].

Nanoparticles (NPs) are the ideal form for targeted and sustained drug delivery to the lungs. This is because drugs administered in their free form (such as through nebulization or aerosolization) to the lungs are unable to achieve sustained release, and NPs can achieve better deposition densities (thus lowering the required dosage of a drug). NPs, when employed as a drug delivery vehicle, have drastically improved pharmacokinetic effects that are able to achieve sustained drug release [26]. By utilizing an NP drug delivery vehicle, a sustained and controlled release can be both achieved and modified by modulating both the drug environment and release profile. Furthermore, currently used drug delivery methods for the lungs, especially in pwCF, present challenges that can be overcome with the addition of a vehicle. An example of this, the TOBI^®^ PODHALER^®^, is discussed later in this paper. Even with the improvements in utilizing an NP as a drug delivery vehicle, there remain challenges to effectively delivering a drug into the lungs in pwCF: these challenges include where the NPs deposit in the lungs, how the NPs interact with the pulmonary mucosal layer and/or biofilms, the cell selectivity, and the patient adherence.

### 3.1. Pulmonary Deposition

The key factor determining where and how particles deposit in the lungs is size. Particles greater than 3 μm deposit in the upper airway (larger primary and secondary bronchi), particles between 0.5 μm and 3 μm deposit deeper in the airway (bronchi and tertiary bronchioles), particles less than 0.5 μm deposit in the alveoli, and particles much less than 0.5 μm are exhaled [27]. There is some disagreement in the literature regarding the fate of the smallest inhaled NPs for the lungs: some studies state that particles between 0.1 and 1 μm will not all deposit, with some small enough to be exhaled; however, particles less than 0.1 μm are paradoxically deposited, though why the smallest NPs deposit instead of being exhaled has not been explained [28]. Notably, many of these studies make no reference to density, so it is unclear if size alone or both size and mass are the critical factors in determining where particles deposit. Regardless, pulmonary deposition is an extremely important consideration when designing an NP for drug delivery. Depending on the target site for drug delivery, NP size can be tuned to ensure an appropriate delivery location, though most often particles are designed to be as small as possible to ensure an even distribution throughout the lungs. Despite this general technique, it should be noted that smaller particles aggregate together more than larger particles, making inhalation and proper deposition of the NPs more difficult [27]. A schematic showing where NPs deposit in the lungs based on size can be seen in Figure 3.

It should be noted that the largest caveat with previous research on pulmonary deposition is that studies have been conducted in healthy individuals with fully functioning airways [28]. In pwCF, it is common to have clogged airways caused by the increased viscosity of the mucus, along with mucus plugs that can fully block off sections of the airway [27,28]. For example, the forced expiratory volume in the first second (FEV_1_) percent predicted (FEV_1_pp) is a percentage that measures what percentage of a person’s lung capacity they can exhale in 1 s and is a good measure for pulmonary function. In healthy adults, an FEV_1_ pp of >80% is considered normal [30], but in pwCF an FEV_1_ of just over 50% is average for patients over the age of 30 [31]. These conditions result in varying deposition profiles in pwCF, where deposition profiles are almost certainly patient-specific [28]. Despite this caveat, smaller particles remain the ideal choice to achieve sufficient pulmonary deposition everywhere in the lungs in pwCF. The reason that deposition must be everywhere is because lung infections in pwCF seem unlikely to be specifically targeted through an inhalation of NPs.

### 3.2. Mucus Penetration

As stated previously, mucus is a gel containing pores and acts as a physical barrier between any inhaled debris and the epithelial cells. There are two options to consider when discussing the mucus layer as a barrier to drug delivery: the mucus as the target for drug delivery, or cells beneath the mucus as the targets for drug delivery. If the target for drug delivery is in the mucosal layer itself, then particles must be designed to adhere to or inside of the mucosal layer, whereas if the target is the cells beneath the mucosal layer, then particles must be designed to not get caught in the mucosal layer.

The two key properties that must be considered when discussing mucus penetration is the pore size in the mucosal barrier and the other solids present in the mucosal layer. The pore size of healthy mucus ranges from approximately 200 to 1000 nm, but in pwCF it is possibly less than 100 nm. This means particles larger than 100 nm will be trapped in the mucus, while particles smaller than 100 nm are able to diffuse through [28,32]. The second key property to consider is the charge of the mucosal layer. Mucins make up the majority of solids in the mucosal layer and are heavily glycosylated, which gives mucins a large negative charge. This is because the terminal sugar of the glycans is often a sialic acid, which has a negatively charged carboxyl group [33]. There is also a sizable portion of free DNA in the mucus from immune cells [34]. This negative charge means that any particles with a positive charge are likely to become trapped, and any particles with a negative charge are likely to be repelled. Particles are thus most likely to diffuse through the mucus layer when they are neutrally charged. Additionally, particles should also be hydrophilic because the mucosal layer, even though it is dehydrated, is still primarily composed of water [28,32,35]. However, if the particles are meant to become trapped in the mucosal layer, then the particles should be designed to be positively charged and potentially hydrophobic. NPs can thus be tuned to be hydrophilic or hydrophobic and have varying surface charges depending on whether the particles are designed to sit in or on the mucosal layer, or are intended to bypass the mucosal layer.

### 3.3. Biofilm Penetration

Infection is the primary problem facing pwCF, *Pseudomonas Aeruginosa* (*P. aeruginosa*) and *Staphylococcus aureus* (*S. aureus*) being the most common bacteria that form these infections [36]. These specific bacteria can persist longer than most other bacteria because of their ability to form biofilms, which greatly enhances survival compared to other planktonic bacteria. Notably, the minimum inhibitory concentration (MIC) for many biofilms has been noted to be approximately 100×–1000× larger than for planktonic bacteria. Biofilms also present the ability to spread effectively; in pwCF, once a biofilm grows large enough, small sections of the biofilm will break off and spread throughout the lungs. As biofilms are specifically created to defend bacteria against the outside environment, administration of purely inhaled drugs is often not sufficiently effective [28]. Therefore, in order for an NP to be effective against biofilms, it likely needs to penetrate the negatively charged biofilm, which points to neutrally charged hydrophilic materials as the best-suited delivery systems [37,38].

### 3.4. Cell Selectivity

In designing an NP for drug delivery, cell selectivity is less of a barrier than the previous barriers (e.g., size and biofilm and mucus penetration). This is because cell selectivity is very important to consider when attempting gene or protein delivery but is less notable when attempting to deliver drugs which typically target bacterial elimination. The most important consideration for cell selectivity is how the NP will be designed to go to specific cells and not others. Oftentimes, cell selectivity is achieved by conjugating an NP with a specific protein on the surface, but this also comes with risks. The protein may be degraded or removed by the time the NP passes through the mucosal layer, reducing the potential to interact with cells lower than the epithelium, which is the target of many gene therapies [28]. In any case, in order for an NP to achieve cell selectivity in pwCF, it must be able to have proteins or other cell-specific targeting molecules conjugated to its surface while still being able to penetrate the abnormal mucosal layer.

### 3.5. Optimal Properties

All NPs should be designed to be as small as possible to achieve optimal pulmonary deposition while preventing particle aggregation. However, beyond that, the optimal properties for an NP are not rigid. Instead, NP properties are dependent on both the drug being delivered and where the drug is being delivered to. For example, to deliver Pulmozyme, a mucus-thinning drug for pwCF that works by breaking up the free DNA in the mucosal layer, the NP should not be hydrophilic and neutrally charged, as bypassing the mucosal layer entirely would not make sense. Instead, a successfully designed NP should stick on/in the mucosal layer. In comparison, when delivering a drug directly to the epithelial cells in the lungs or to bacteria in a biofilm, a neutrally charged and hydrophilic NP would be ideal; here, the NP should penetrate these layers, and this configuration would not lead to entrapment in the biofilm or mucus. Therefore, the only rigid parameter in NP fabrication is size, and all other parameters may be highly tuned to create the best-suited NP for the target goal. The tunable properties are all material-specific, thus making material selection the most important choice when designing an NP for pwCF. This is because material choice is the determining factor in how an NP may be tuned to change important properties, such as surface charge, functionalization, size, and so on. For this reason, the rest of this paper will be spent discussing three major classes of NPs: metallic NPs, polymeric NPs, and lipid-based NPs.

## 4. Metallic Nanoparticles

Metallics are an attractive material choice for NPs, as they are one of the longest-studied material choices (in humans) and have already achieved clinical benefits when used in implantable devices [39]. Additionally, metallic NPs possess unique properties that make them extremely attractive as material choices, with further subgroups including silver, gold, and iron oxide NPs.

### 4.1. Silver Nanoparticles

Silver NPs often exist in a +1 oxidation state (+2 or 3 is also possible) and are fabricated in size ranges of 10–200 nm [40]. They can also be made into various shapes, such as spheres, shells, rods, ribbons, and cubes [38]. Notably, silver is an attractive material choice for NPs because silver is innately bactericidal, which is achieved by a variety of methods [41]. Though all the mechanisms responsible for this are not fully understood, it is known that one way in which silver is bactericidal is through the generation of reactive oxygen species (ROS). Briefly, when silver is exposed to an oxidizing agent (commonly O_2_), it is oxidized to become a silver cation (most commonly, Ag^+^). This oxidation reaction commonly creates ROS such as O_2_^−^ [41]. Note that these ROS are extremely detrimental to all cells because of their ability to disrupt cellular membranes and damage DNA [42,43]. Therefore, while bacterial cells are more susceptible to this damage than eukaryotic cells, ROS can also be harmful to eukaryotic cells. If silver is chosen as an NP source for pwCF, this cytotoxicity must be carefully balanced to create an effective therapy.

### 4.2. Gold Nanoparticles

Gold NPs generally exist in a +3 oxidation state (+1, 2, 5, 7, and even −1 are also possible) and have been fabricated in sizes ranging from 5 to 100 nm [44] and shapes such as spheres, rods, shells, cages, and stars [38]. Gold is attractive as a material choice for NPs because it is considerably more bioinert than other metals. Studies have shown it to be non-cytotoxic, with no ROS production, a common occurrence with other metals [45]. However, it should be pointed out that one study noted the presence of anti-gold and anti-silver antibodies in rabbits, demonstrating the possibility that while these cells are non-cytotoxic, they may still be immunogenic and have been known to increase proinflammatory cytokines [46]. If gold NPs are proinflammatory, then their use as NPs for pwCF could potentially be excluded because inflammation is a serious problem in pwCF and adding to it can be very dangerous.

### 4.3. Iron Oxide Nanoparticles

Iron oxide NPs have three commonly used states due to the various oxidation states that iron can occupy: maghemite (γ-Fe_2_^3+^O_3_), hematite (α-Fe_2_^3+^O_3_), and magnetite (Fe_3_^2.67+^O_4_) [47]. Iron oxides are interesting as NPs, as they are superparamagnetic and thus are often called superparamagnetic iron oxide NPs (SPIONs). Superparamagnetic here refers to the fact that the magnetic moments of the atoms present in the NPs all point in the same direction. When not in a magnetic field, ferromagnetic NPs have magnetic moments in many different directions, often differentiated by grain boundaries, while SPIONs are small enough and magnetic enough to have only a single directional magnetic moment. When in the presence of a magnetic field, both ferromagnetic NPs and SPIONs align their magnetic moments in the direction of the magnetic field, but when removed from the magnetic field, SPIONs instantly revert to their original magnetic moment orientation, while ferromagnetic NPs are much slower and may retain some of their magnetization from the magnetic field, as they slowly dissipate energy received from the magnetic field [48]. A schematic of this process can be seen in Figure 4.

This property of SPIONs is interesting because it opens two potential therapies in pwCF. The first is hyperthermia treatments. When in the presence of a rapidly oscillating magnetic field, the magnetic moments of SPIONs can rapidly change and the resulting energy from this oscillation is released as heat. This allows for potential adoption of inhaled SPIONs, followed by physical placement of the patient in a rapidly oscillating magnetic field to internally heat their lungs, which may help in mucosal clearance through loosening. The second is to use SPIONs as “metallic nano-knives”. Once inhaled, the SPIONs could be moved with an external magnetic field to forcibly push through mucus in the lungs, thus loosening it to assist with clearance [49]. While such a use for SPIONs has been proposed, in reality it is not feasible given the extremely complex geometry of the lungs.

### 4.4. Metallic Nanoparticle Functionalization

Metallic NPs are often functionalized with either the drug to be delivered or other materials to reduce the immunogenic effect of the NPs. Conjugation to the surface of metallic NPs is achieved through chemical adsorption and, as most drugs will not spontaneously adsorb onto metallic surfaces, these NPs must adsorb an additional component that drugs can then covalently bind to. For silver and gold NPs, thiols and disulfides are commonly used, as they spontaneously attach to these metallic NPs; other common choices include phosphines, amine, nitriles, carboxylic acid, and citrate. For iron oxide NPs, it is more common to use phosphonates, silane, and carboxylic acid, though amines and thiols are also occasionally used. Conjugation of metallic NPs with functional groups also serves to increase the stability of these NPs, both in solution and by preventing aggregation once in the lungs, through both steric hinderance and electric repulsion [50].

Functionalization of metallic NPs with other materials is also incredibly important because functionalization can drastically alter the body’s response. In human serum, it is well known that the biological identity of an NP is discerned by the protein corona that covers the NP. However, the makeup of this corona can be easily altered depending on the molecules on the surface of the NP. A metallic NP coated in polyethylene glycol (PEG) will not have the same immunogenic response as a metallic NP coated with citrate. For this reason, it is possible to modulate the non-ideal immunogenic responses exhibited by these metallic NPs [51]. Table 1 summarizes some recent examples of these metallic NPs.

## 5. Polymeric Nanoparticles

Polymeric NPs are composed of polymers. Polymers are defined as the multiple repetition of atoms or groups of atoms (monomers) in amounts such that the properties associated with the chain do not vary substantially with the addition or subtraction of some of the repeating units [60]. Polymers can be divided into three types: natural, synthetic, and semisynthetic. Natural polymers arise from organisms, and commonly used organic polymers for NPs include gelatin, alginate, cellulose, and chitosan. Synthetic polymers are man-made, and commonly used synthetic polymers for NPs include PEG, polycaprolactone, polylactic acid, and countless more. The final type of polymer is semisynthetic polymers, which are any natural polymers that are modified through various chemical processes to alter their properties [61]. When choosing a polymer, while the material itself is important, so is the size of the polymer. For any given polymer, chains can be hundreds to thousands of monomers long, and these varying chain lengths can provide very different properties for a given polymer, especially in the body. This chain length can alter many different properties, such as viscosity, hydrophobicity, crystallinity, melting temperature, and many more [60]. 

### 5.1. Polyethylene Glycol

PEG is synthesized using a base-catalyzed polymerization reaction of ethylene oxide. It has an average bulk density of 1.2 g/cm^3^, and the molecular weight can vary from 200 Da to 4,000,000 Da [62]. However, once above a molecular weight of 100,000 Da, PEG is referred to as polyethylene oxide (PEO). PEO and PEG have the exact same molecular structure but different properties due to the change in molecular weight. PEG is unique for NPs, as it is not commonly used as the sole material choice but is instead applied in small amounts in conjugation with other materials. An example of this is in the COVID-19 mRNA vaccines produced by Pfizer/BioNTech and Moderna that use liposomes with PEG conjugated to the surface [63]. A figure depicting PEG’s structure can be seen in Figure 5.

PEG has many key properties that make it useful as a conjugate for NPs. The first is its hydrophilicity and neutral surface charge. PEG is extremely water-soluble and often used to increase the solubility of other materials in polar solvents. This can be helpful in the fabrication of NPs, especially in pwCF, as a conjugate to help particles move in the mucosal layer. As stated previously, the mucosal layer is largely negative, so materials that are hydrophilic and neutral have reduced interaction with the mucosal layer. As the molecular weight of PEG increases, so too does the solubility and hydrophilicity, due to increased hydrogen bonding with water. Further, conjugating NPs with PEG increases steric hinderance, preventing particle aggregation, increasing deposition efficiency, and improving stability. PEG decreases non-specific protein interaction both in the mucosal layer and in the bloodstream, which, as stated previously, can decrease non-specific protein binding to NP surfaces and improve the immunogenic response. It is also easy to functionalize and can be terminated with many different functional groups, such as thiols, acrylates, amines, and benzoates; this is useful in both tuning the surface properties and conjugating further molecules to PEG [63].

Finally, while PEG has positive immunogenic effects and serves to decrease immunogenic reactions, it does present some drawbacks. Despite extensive research on PEG, there are reports of anti-PEG antibodies and pseudo-allergic effects due to PEG NPs. These results may indicate that PEG can increase inflammation if delivered directly to the lungs, which can be dangerous in pwCF. While pseudo-allergic effects have not been conclusively attributed to PEG’s inclusion in NPs, it is still a risk and requires additional research [63].

### 5.2. Chitosan

Chitosan is a derivative of the naturally occurring organic polymer chitin, a polysaccharide commonly found in fungi as well as the exoskeletons of crustaceans and insects [65]. Chitin is a polymer of N-Acetylglucosamine, which is itself a derivative of glucose. To create chitosan, chitin chains undergo a deacetylation reaction with a base (commonly, sodium hydroxide) at high temperatures to remove the 2′ acetyl group [66], and the monomers become glucosamine (a monosaccharide commonly found in human cartilage) [66,67]. It should be noted that this deacetylation of chitosan to chitin is not 100% efficient and can vary depending on the length of the reaction and the base used. Additionally, different degrees of deacetylation can result in varying polymer properties; thus, more or less acetylation may be desirable in certain instances. The molecular weight of chitosan can range from 4700 Da to 375,000 Da, and the bulk density of chitosan is approximately 0.15–0.38 g/cm^3^ [68]. Chitosan also has a pKa of ~6.5, indicating that it is soluble in slightly acidic conditions [69]. Thorough depictions of the structure and functionalization of chitosan can be seen in Figure 6 and Figure 7.

The functionalization of chitosan is achievable through a variety of methods (Figure 7). This allows for drugs to easily be conjugated to chitosan and for properties to be highly tunable. An example of this is the quaternization of the primary amine group, which improves the water-solubility of chitosan [70]. Furthermore, chitosan has four important material properties that make it attractive as an NP for pwCF. First, chitosan is a mucoadhesive. While penetrating the mucosal layer is often the goal for drug delivery vehicles in pwCF, it can also be advantageous for drugs to be delivered directly to the mucosal layer and then remain there for sustained drug release. Second, chitosan is a natural antioxidant that can alleviate oxidative stress induced by bacterial infections. The hydroxyl and amine groups present on chitosan can deactivate free radicals, and the less acetylation present on the chitosan, the better it is at scavenging free radicals. Third, chitosan is inherently antimicrobial. It can increase the cellular permeability of bacteria through amine interactions with carboxylate present in cell membranes. Fourth, and finally, chitosan is an anti-inflammatory material that reduces nitric oxide production that modulates pro-inflammatory mediators. This can be very important in pwCF, who generally experience chronic lung inflammation that may further exacerbate lung damage [71].

While chitosan has been approved by the FDA for would healing and for ingestion, it has not been approved as a drug delivery vehicle. Initial animal-model studies indicate that chitosan appears to be non-cytotoxic and mostly bioinert [71]. This is reasonable, as chitosan is similar in structure to sugars used in the human body, as stated previously.

### 5.3. Poly Lactic-co-Glycolic Acid

Poly Lactic-co-Glycolic Acid (PLGA) is a synthetic copolymer composed of lactic acid and glycolic acid monomers (Figure 8) [72]. The latter figure, however, is not an accurate representation of PLGA. PLGA is defined with the word *co* (Poly Lactic-co-Glycolic Acid), which, according to the International Union of Pure and Applied Chemistry (IUPAC)’s naming conventions, means that it has an undefined polymeric structure [73]. The image makes it appear as though lactic acid (LA) forms by itself and then a large LA chain joins to a large glycolic acid (GA) chain, which is not necessarily the case. PLGA is instead simply a copolymer with some unspecified structure that cannot be characterized with other naming conventions (i.e., not a block, random, alternative, statistical, etc.). A figure depicting PLGA’s structure can be seen in Figure 8.

High-molecular-weight (MW) PLGA (>10 kDa) is synthesized using a catalyzed ring-opening polymerization reaction, while low-MW PLGA (<10 kDa) can be synthesized using the same reaction either without a catalyst or by using a direct polycondensation reaction [72,74]. Research is currently focused on synthesizing PLGA in microorganisms such as *Escherichia coli* [75]. Generally, it has a MW between 5 and 40 kDA but can also stretch up to 100 kDa [72,74]. Since PLGA is a copolymer, varying the ratio of lactic acid (LA) to glycolic acid (GA) can drastically alter the properties of PLGA. The most commonly used ratios of LA to GA are 50:50 and 75:25 [76].

Notably, PLGA can also be tuned because of the chiral center exhibited in the LA portion of the structure. LA has a chiral center caused by the methyl group, meaning that PLGA can be formed using D, L, or DL isomers of LA. The L isomer of LA has an extremely crystalline and ordered structure, while the D isomer of LA is completely amorphous. GA, having no chiral center, is always crystalline. Most PLGA copolymers use a racemic mixture of DL LA isomers which, when combined with crystalline GA, results in an amorphous structure of PLGA [74].

Extensively studied, PLGA is desirable as a material for drug delivery due to its biodegradable body and high tunability to alter the degradation rate [77]. It has three different ways in which the properties can be altered to meet desired NP properties, while all other polymers discussed in this section generally only have one (that one being MW). Furthermore, the degradation of PLGA is significantly slower than that of lipid-based and organic NPs, making it an excellent material choice to achieve sustained drug release [78]. PLGA can be altered in the following three ways: MW, the LA-to-GA ratio, and the type of LA used.

PLGA is broken down in the body, through the hydrolysis of its ester bonds, into its constituent components, LA and GA, both of which are natural byproducts [72,74,77]. LA is the major product of anaerobic cellular respiration and is easily tolerated by the body. It can be metabolized by cells, where it is either broken down into CO_2_ and H_2_O, converted back into glucose, or removed from the body via the kidneys [79], thus explaining why PLGA has an extremely low immunogenic effect. While toxicology reports state that there may be a slightly toxic effect in tissue directly exposed to PLGA NPs, it is unclear if this is because the NPs are composed of PLGA or because they are NPs [77].

The rate at which an NP is broken down determines how sustained and controlled the drug delivery rate is. For PLGA, this can be achieved in one of three ways, as stated previously. First, the MW can be altered, where the larger the polymer chains become, the longer it takes for the polymer to be degraded, thus increasing how long the NPs may remain. Second, altering the LA-to-GA ratio has many effects and allows for an extremely tunable polymer. In terms of hydrophilicity, LA is more hydrophobic than GA because of the methyl group causing steric hindrance, but both molecules are still hydrophilic. By altering the ratio of LA to GA, the hydrophilicity can be altered, where increased hydrophilicity increases the degradation rate of PLGA. This tunable hydrophilicity can also be helpful in altering the surface properties of PLGA NPs. Finally, the type of LA isomer used in the polymer can be changed. As stated previously, LA has two different isomers with varying degrees of crystallinity. The D isomer degrades much faster than the L isomer because the D isomer is amorphous while the L isomer is crystalline, and this dense packing helps protect it from hydrolysis. In PLGA, a mixture of DL isomers is used, allowing for a degradation rate slower than that of D isomers but faster than that of L isomers [74]. However, these results are not fully understood and require additional research [77].

The functionalization of PLGA is well understood and is commonly completed with PEG, specific targeting ligands, bilayer lipids, monolayer lipids (through physical adsorption), and other surfactants. Since PLGA can be conjugated with PEG on the surface, anything that PEG can be functionalized with, PLGA can also be functionalized with. PLGA can also chemically adsorb amide, thioether, triazole, thiol, aldehyde, and maleimide groups, which can result in further functionalization [80,81]. As with chitosan and PEG, PLGA exhibits high degrees of functionalization.

### 5.4. Alginate

Alginate is an organic, hydrophilic, anionic polysaccharide generally extracted from brown marine algae, though it may also be extracted from bacteria. It is extracted as alginic acid before being treated with an alkali solution to form the salt of an alginic acid: alginate [82]. The most common form for alginate to take is sodium alginate. Alginate is a copolymer composed of D-mannuronic acid and L-guluronic acid. D-mannuronic acid is connected to other D-mannuronic acid monomers or L-guluronic acid monomers through a β-(1,4) linkage, and L-guluronic acid monomers are connected with an α-(1,4) linkage [83]. A sketch of the molecular structure of guluronic acid and mannuronic acid groups can be seen in Figure 9.

Alginate has a MW generally between 30 Da and 400 kDa [84]. Many of alginate’s properties arise from how it is crosslinked to form a hydrogel, a solid structure that is extremely hydrated and is capable of both swelling and degrading [85]. Alginate is crosslinked using divalent or trivalent cations, though it is most common to use Ca^2+^ due to demonstrated clinical safety and efficacy [86]. The process through which alginate is crosslinked is described as the egg-box model (Figure 10) [84]. In this model, separate antiparallel strands (or even the same continuous strand) crosslink together through the coordination of divalent or trivalent ions, which interact with the negatively charged carboxyl groups and other oxygen atoms to form a stable complex. It is generally considered that only the guluronic acid monomers can participate in this crosslinking/gelation process, making the ratio of guluronic to mannuronic acid a key property in alginate [82,83,84,86]. A depiction of the egg-box model can be seen in Figure 10.

Alginate possess various properties that make it desirable as an NP for pwCF. First, alginate is an incredibly strong mucoadhesive polymer and is thought to be stronger than others, including polystyrene, chitosan, and cellulose [82]. It makes sense that alginate is a strong mucoadhesive because alginate is also the polymer that makes up the majority of a *Pseudomonas aeruginosa* biofilm [88]. This can be incredibly useful for mucosal layer or biofilm attachment in the lungs of pwCF. Second, alginate’s gelation is unique compared to other saccharides that gel because it can occur at room temperature [82]. This means that alginate is suitable for encapsulating drugs that are sensitive to higher temperatures, such as proteins and nucleic acids.

Through modification of the molecular weight, the rigidity of alginate NPs can be adjusted (e.g., increased MW will decrease the rigidity, changing the erosion time of alginate NPs). Further, pore size, which effects drug elution, can be affected by the G:M ratio (more G leads to a more open pore structure with less shrinkage). Overall, a low MW and a high G:M ratio is considered the strongest combination and offers optimal protection of drugs for delivery. The way in which alginate cations are exposed to alginate to initiate crosslinking also leads to various properties. Internal gelation leads to more homogenous gels with a uniform pore size and faster drug release, while external gelation crosslinks the surface so that drug release is difficult and there is a delay in diffusion [83].

While some studies demonstrate that alginate is generally bioinert, other studies have shown that it can possibly be cytotoxic. It may be that this cytotoxicity only occurs because purification methods are inadequate and certain alginates leave cytotoxic impurities behind [82]. Notably, alginate is non-degradable in humans because there is no enzyme to digest it (alginase); however, alginate can be dissolved through the release of cations. It is generally excreted through renal clearance, though some may accumulate in the body without posing any danger [82]. Table 2, shown below, summarizes a few recently used polymeric NPs discussed in this section.

## 6. Lipid-Based Nanoparticles

Organic molecules are classified into four broad groups: (i) proteins, composed of amino acids and linked by peptide bonds; (ii) carbohydrates, composed of monosaccharides and linked by glycosidic bonds; (iii) nucleic acids, composed of nucleotides and linked by phosphodiester bonds; and (iv) lipids which are much less strictly categorized and are simply categorized as any non-polar organic molecule [98]. Typical examples of lipids include fatty acids, triglycerides, steroids, waxes, and phospholipids. Lipids have many uses in organic systems, but their major function is as the structural component of cellular membranes. Phospholipids, specifically, are the major components of all eukaryotic cellular membranes and are the structural elements used in all lipid-based NPs. Their important properties are caused by their polar head groups, combined with their hydrophobic tails [99]. Steroids, another category of lipids, are also important structural elements used in both cellular membranes and lipid-based NPs.

### 6.1. Liposome and Micelle Formation

Lipid-based NPs are characteristically formed into one of two structures (micelles or liposomes), the key difference being the presence or lack of a bilayer sheet of phospholipids. Micelles are formed by dissolving phospholipids in a solvent (commonly water), and when enough phospholipids are put into solution, they spontaneously coalesce to form a micelle due to their amphiphilic nature; the polar head group faces out into the solvent, and the hydrophobic tails face in to exclude the solvent from the center of the micelle. This is an entropically driven process, which may not appear clear at first glance, as micelles are highly ordered structures and appear to decrease the entropy of a solution. However, the formation of micelles increases entropy because, when the hydrophobic tails are exposed directly to a solution (water in this case), the solution forms a clathrate-like structure/shell/cage around the hydrophobic tails by hydrogen bonding to other water molecules. This is an entropically unfavorable process, as these clathrate shells exhibit an increased number of hydrogen bonds and thus reduced mobility when compared to free water molecules. Through micelle formation, these clathrate shells are released, thus increasing entropy. This spontaneous formation makes the encapsulation of hydrophobic drugs easy, as micelles will form around the drugs (encapsulation) and decrease entropy. Notably, micelles may also have micelles within themselves. This allows the micelles to encapsulate hydrophilic substances as well, as the smaller micelles within larger micelles will have polar head groups facing in to encapsulate the hydrophilic substance and tails facing out and touching the larger micelles’ hydrophobic tails [100].

Liposome formation occurs in the same manner, with the only difference being that instead of hydrophobic tails pointing into the shell to exclude any solvent from the micelle center, liposomes form a bilayer of phospholipids, identical to the structure of cell membranes. This formation of liposomes is also entropically driven, for the same reason that micelle formation is entropically driven. What determines whether a phospholipid forms a micelle or a liposome (or one of the many other possible configurations not discussed here) is the size of the hydrophobic tails along with the size of the polar head group (and the ratio of the two) [101]. Liposomes are also able to have multiple layers of bilayer sheets within them, and when they do, they are called multilamellar liposomes.

### 6.2. Types of Lipid-Based Nanoparticles

There are four different types of lipid-based NPs (Figure 11) that differ in construction and have varying properties. Liposomes are made using an organic solvent and generally have a low encapsulation efficiency. They were notably the first fabricated lipid-based NPs but soon evolved into solid lipid nanoparticles (SLNPs), which use a surface-active agent (often a phospholipid) and a solid crystalline lipid to improve stability and loading capacity. However, SLNPs have a poor shelf life because the solid crystalline lipid will slowly expel the drug encapsulated within it. Alternatively, nanostructured lipid carries (NLCs), evolved from SLNPs, possess the solid crystalline lipid also incorporated with a liquid-phase lipid, which prevents drug expulsion and improves shelf life. Despite this, shelf life remains a major problem for all lipid-based NPs, as they generally cannot be fabricated over 3–4 weeks before administration due to instability in storage [102].

### 6.3. Phospholipid Structure and Functionalization

Lipid-based NPs are generally modified to create a specific drug release profile. This release of the drug is generally accomplished through destabilizing the lipid-based NPs under certain conditions. The first way this can be accomplished is through modification of the polar head group. Phospholipids all have the same general structure of a polar head group connected to a glycerol backbone through a phosphate group, and then two fatty acids. The polar head group may be tuned to be cationic, anionic, zwitterionic (neutral), or ionizable cationic [103]. Examples of possible head group modifications can be seen in Figure 10. In naturally occurring phospholipids in the cell membrane, the polar head group is generally phosphatidylcholine, which is zwitterionic. However, for creating a lipid-based NP, these head groups can be modified depending on the desired encapsulation molecule. Cationic head groups are often best for encapsulating nucleic acids because of the interactions between the positive head groups and the negatively charged nucleic acids. These also increase interactions to support cellular membrane uptake. However, because cationic head groups are not found in nature, they are generally considered cytotoxic and highly immunogenic. Anionic head groups are generally not used because they are difficult to produce, exhibit increased cytotoxicity, and induce pseudo-allergic effects [103].

The most commonly used head groups are zwitterionic and ionizable cationic groups. Zwitterionic groups have a decreased interaction with proteins, which prevents non-specific protein aggregation but also leads to non-specific drug release. Zwitterionic groups also improve cellular uptake and are non-toxic, as they mimic organically found phospholipids (e.g., phosphatidylcholine). Ionizable cationic groups are designed so that the head groups only become positive when there is a pH decrease, such as inside a cell, which can lead to greatly increased drug release with cells, which is especially important for gene therapies. A figure depicting phospholipid head modifications can be seen in Figure 12.

Beyond phospholipid head group modifications, alterations can also be made to the fatty acid tails. Naturally occurring phospholipids in cellular membranes have fatty acids ranging from 14 to 20 carbons long and generally possess one unsaturated and one saturated fatty acid. One of the most important parameters for phospholipids is the gel-phase transition temperature (T_C_), which is a way of measuring how fluid the membrane is. When phospholipids are above this temperature, they are a liquid and generally considered less stable. When they are below this temperature, they exhibit a gel-like phase. The chain length of phospholipids can modulate T_C_. Increasing the chain length decreases the fluidity of lipid-based NPs and increases T_C_ because of increased interactions between hydrophobic tails. Decreasing chain length has the opposite effect and will increase the fluidity of the NPs. Fatty acid chains can also be either saturated or unsaturated. Fully saturated bonds decrease the fluidity of lipid-based NPs because the phospholipids can achieve tighter packing with less steric hinderance, thus increasing T_C_. Unsaturated fatty acids have the opposite effect and increase the fluidity because of increased steric hinderance. Furthermore, altering the chain length and degree of hydrogenation (saturated vs. unsaturated) may affect the layer thickness, which can be very important for encapsulating hydrophobic drugs within the hydrophobic portion of the liposome [105].

A final way to modulate membrane fluidity is with cholesterol. Cholesterol is another lipid commonly found in cellular membranes and is the most impactful way to modulate membrane fluidity in lipid-based NPs. Cholesterol increases the fluidity of gel-phase lipids and decreases the fluidity of liquid lipids. In many formulations, cholesterol makes up 30 mol% of lipid-based NPs [104].

Lipid-based NPs are generally used to encapsulate drugs, and the polar head groups and fatty acid chains are highly tunable to alter the stability of lipid-based NPs; however, lipid-based NPs can also be highly functionalized to include other molecules at their surface. This can be extremely effective in trying to create an NP with either increased cell interactions by conjugating an antibody to the surface or decreasing cell interactions by coating with PEG [106]. A schematic showing lipid-NP functionalization and encapsulation can be seen in Figure 13.

Lipid-based NPs can have varying degrees of immunogenicity depending on how the phospholipid has been tuned and whether other excipients, such as cholesterol, are used. Generally, it is the case that the more closely the phospholipids align with naturally occurring phospholipids, the less immunogenic they are, whereas the more they deviate from naturally occurring phospholipids, the more immunogenic they are. This is especially true for cationic and anionic polar head groups. NP size is also important, as it has been shown that lipid-based NPs exhibit size-dependent cytokine activation, with smaller NPs causing less activation than larger ones. Lipid-based NPs may also be complement cascade activators and potentially cause hypersensitivity reactions. Note that it remains unclear if this is caused solely by the phospholipids. The type of hydrophobic chain chosen (e.g., the length and degree of saturation) is unknown and likely does not contribute to any immunogenic effect [107]. Table 3, shown below, summarizes a few recently studied lipid NPs.

## 7. Current FDA-Approved Inhalable Nanoparticles

Currently, there are three types of inhaled therapeutics available for pwCF. The most common inhaled therapeutics are antibiotics such as tobramycin, aztreonam, amikacin, and colistimethate. These can all be used to treat *P. aeruginosa*, one of the most common lung infections in pwCF, along with other Gram-negative bacteria. Additional inhaled therapeutics include bronchodilators to improve mucosal clearance and help treat inflammation in the lungs, including albuterol, ipratropium bromide, terbutaline sulfate, and an inhaled mucus-thinning agent known as dornase alfa, to improve mucosal clearance [114]. Of all the above inhaled therapeutics for pwCF, only two, both antibiotics, use NP formulations while the rest use free drug formulations with other excipients to improve delivery or shelf life. These antibiotics are tobramycin, which can be inhaled freely or in an NP formulation with the TOBI^®^ PODHALER^®^, and amikacin, which can be inhaled with ARIKAYCE^®^.

### 7.1. TOBI^®^ PODHALER^®^

The TOBI^®^ PODHALER^®^ from Novartis is a lipid-based NP that first achieved FDA approval in March 2013 [115]. The primary antibiotic used is tobramycin, an aminoglycoside, which first received FDA approval in 1975 to combat *P. aeruginosa* infections in pwCF [116]. The TOBI^®^ PODHALER^®^ uses PULMOSPHERE^®^ technology to create a porous lipid-based NP composed primarily of 1,2-Distearoyl-sn-glycero-3-phosphocholine (DSPC), with an encapsulation efficiency of 90–95% *w*/*w* [115]. These particles are created through a process of spray drying, where, first, tobramycin and DSPC are emulsified in an oil-and-water solution which causes DSPC micelles to form around oil droplets, with tobramycin dissolved in the water. This emulsion then undergoes spray drying, and, as the water evaporates from the droplets, the micelles move closer to the surface of the droplets and tobramycin moves towards the center. Finally, as the oil inside the micelles and the last of the water evaporates, highly porous NPs with tobramycin inside of them are created. These NPs have a median geometric diameter between 1.7 and 2.7 μm or 1700 and 2700 nm, with a mass median aerodynamic diameter of less than 4 μm. The surface area is ~90% DSPC, which serves to lower the surface energy of the NPs and prevents conglomeration, which improves delivery to the lungs [117].

Three phase III clinical trials were performed to understand the efficacy and safety of the TOBI^®^ PODHALER^®^. The first clinical trial, EAGER, directly measured the safety, efficacy, and convenience of the TOBI^®^ PODHALER^®^/Tobramycin Inhalation Powder (TIP) versus TOBI^®^/Tobramycin Inhalation Solution (TIS) [118]. TIS is the original formulation for inhaled tobramycin and is a nebulized solution of tobramycin in solution, instead of tobramycin in NPs. In EAGER, 553 pwCF were assigned 3:2 to TIP (112 mg tobramycin twice daily) or to TIS (300 mg tobramycin twice daily). Individuals were treated in three cycles, with each cycle consisting of 28 days on the drug and 28 days off. The results of this study showed that the rate of cough suspected to be related to the administered drug was higher for TIP patients than TIS patients (25.3% and 4.3%), and the discontinuation rates for TIP and TIS were similar (26.9% and 18.2%). The forced expiratory volume in 1 s (FEV_1_)% predicted increases from baseline to the end of cycle 3 were comparable, as were the reductions in the *P. aeruginosa* density in patient sputum (log_10_ CFU/g) at the end of cycle 3. Most notably, the time required to administer TIP was 5.6 min on average, but for TIS it was a staggering 19.7 min, and the treatment satisfaction for TIP was greater than that for TIS [118].

The second study, EVOLVE, tested TIP administration in 95 pwCF, aged 6–21, who were randomly assigned to receive TIP twice daily (n = 46) or a placebo (n = 49) for 28 days and then take 28 days off medication. Two more open-label cycles with all patients were then performed after this, following the same cycle of 28 days on and 28 days off. The study concluded that TIP improved FEV_1_% predicted versus placebo (a 13.3% increase) at the end of cycle 1, and similar results were observed in placebo patients at the end of cycle 2, with improvements for all groups being maintained over the course of the study. *P. aeruginosa* density and hospitalizations were reduced in the TIP group compared to the placebo group, and the only adverse event was coughing, which was higher in placebo patients (26.5%) than in TIP patients (13%) [119].

The final study, EDIT, tested TIP administration in 62 pwCF, aged 6–21, who were randomly assigned to receive TIP twice daily (n = 32) or a placebo (n = 30) for 28.5 days and then take 28 days off medication. This study was extremely underpowered due to difficulties with recruiting patients, making conclusions difficult to draw, but the results were mostly promising. The main results were a mean treatment difference between TIP and placebo of 4.4% for absolute change in FEV_1_% predicted (*p* < 0.05) and 5.9% for relative change in FEV_1_% predicted (*p* = 0.148), and a −1.2 log_10_ CFU (*p* = 0.002) in *P. aeruginosa* density in sputum [120].

These studies showed that TIP is as effective as TIS and is well tolerated in pwCF. It also showed a greater satisfaction in pwCF using TIP than in those using TIS, but this is not the only benefit TIP has over TIS [118]. TIP is stored at 25 °C (77 °F); in comparison, TIS must be stored at 2–8 °C (36–46 °F) before being given to a patient and then at 25 °C (77 °F) for a maximum of 4 weeks—the length of a prescription. This demonstrates the improved shelf life of tobramycin NPs vs. free tobramycin. Furthermore, TIS has a dosage of 300 mg of tobramycin, while TIP has a dosage of only 112 mg of tobramycin, thus showing equivalent effects with a lower dosage, possibly because of more efficient lung deposition of the NPs [115,116].

### 7.2. ARIKAYCE^®^

While the TOBI^®^ PODHALER^®^ uses a non-traditional lipid-based NP, ARIKAYCE^®^ uses a liposome to deliver drugs. ARIKAYCE^®^ received FDA accelerated approval in 2018 for the treatment of *Mycobacterium avium* complex lung infections in patients who, after 6 months of multidrug treatment, do not have a negative sputum culture. There is limited safety and efficacy information, so this drug is reserved for patients with limited or no other treatment options [121]. The liposomes are structures with amikacin as the primary antibiotic and the liposome is formed with 1,2-Dipalmitoylphosphatidylcholine (DPPC), along with cholesterol to improve the fluidity [121].

One phase III clinical trial, CONVERT, has been performed to evaluate the safety and efficacy of ARIKAYCE^®^. Patients were selected based on having *Mycobacterium avium* complex lung infections and continuing to have positive sputum cultures after 6 months of standard therapy. Patients were randomly assigned to receive ARIKAYCE^®^ once daily and standard therapy (n = 224) or standard therapy alone (n = 112). The primary endpoint was 3 consecutive months of negative sputum cultures by the sixth month. Of the 224 patients who received ARIKAYCE^®^, 65 (29%) patients achieved this primary endpoint, while only 10 (8.9%) achieved this endpoint with the standard therapy alone. However, adverse respiratory events were reported in 87.4% of patients who received ARIKAYCE^®^ and only 50% of the patients who received standard therapy [122]. These adverse effects were likely not caused by the liposomes but instead by amikacin, though this has not been studied to the best of our knowledge [121].

## 8. Conclusions and Future Directions

This paper intended to discuss Cystic Fibrosis and the material selections for NP-based treatments, most often focused on bacterial infection. While metallic, polymeric, and lipid-based NPs were discussed, key attention was only paid to specific materials within those classes. Furthermore, NPs that use combinations of different materials (such as polymeric NPs coated in lipids or metallic NPs coated in polymers) were left undiscussed. Combining material choices is a way to circumvent many of the negative properties of a specific material, as many interactions between NPs and the body occur at the NP surface, and by coating the NP in a different material, different immunogenic effects can be elicited. In fact, many currently studied NPs do combine two or more material choices, often from different classes. Furthermore, many NPs often include PEG in their formulations, as it is generally considered a safe and effective way to modulate immune interaction.

For the future of NP formulations for CF, it is unclear which material choice will yield the greatest benefits. From this review, lipid-based or polymeric NPs are most likely to have the best outcomes due to higher biocompatibility and tunability. Lipid-based NPs appear to have the best immunogenic properties and their degree of tunability is significantly higher, and better understood, than metallic and polymeric NPs. Furthermore, they can encapsulate both hydrophilic and hydrophobic drugs and the two currently FDA-approved NPs are lipid-based. However, lipid-based NPs have the downside of possessing extremely poor shelf lives, except for the TOBI^®^ PODHALER^®^, which uses a non-standard lipid-based NP. Polymeric NPs, on the other hand, also have very positive immunogenic effects and high tunability and functionalization, as well as a significantly improved shelf life. Polymeric NPs have the complication of being more immunogenic than lipid-based NPs and have yet to receive FDA approval. Metallic NPs do not seem likely to become a major treatment option for CF. These NPs have poor immunogenic effects, though this can be modulated with PEG coating, but it remains unclear why a metal should be chosen over a polymer or lipid. Metallic NPs do have the possibility of hyperthermia treatments with SPIONs, which is an interesting treatment option but seems less likely to succeed than traditional drug treatments. Therefore, it remains clear that polymers and/or lipids are the future of CF NP-based treatment. While lipid-based NPs possess the best immunogenic effects, their poor shelf life and stability makes them less economically and practically feasible than polymeric NPs with much more stable shelf lives.

Finally, patient adherence is a considerable problem when discussing CF that is often overlooked. It has been shown, and can easily be inferred, that patients who do not stick strictly to their medical regimens have worse outcomes [123]. While this finding is not surprising, it is also not surprising that patients do not stick strictly to the intense medical regiments that CF treatment requires. Treatments can take two–three hours daily and provide a serious burden for all patients. Furthermore, since CF is a genetic disease in which symptoms first present at birth, it must be remembered that many patients undergoing CF therapy are children. It can be difficult to explain to them why they must undergo these therapies, and even effective therapies can place tremendous financial and emotional strain on both children and their families [124]. It is also extremely important for children to undergo these therapies because lung damage that occurs early in pwCF can lead to significantly worse outcomes later in life [123]. Any new therapy/drug should be designed with patient adherence in mind. The ease of use and effectiveness must both be maximized so as not to increase the patient burden and increase the likelihood of adherence to the medical regimen.

## Figures and Tables

**Figure 1 biomimetics-09-00574-f001:**
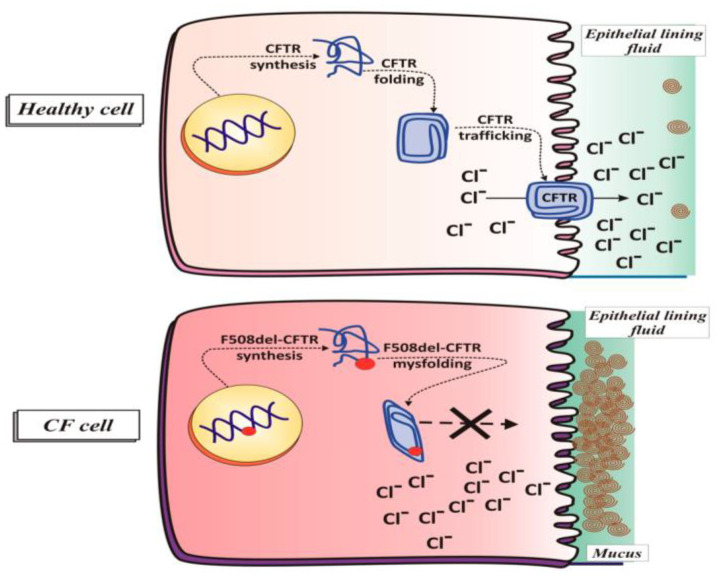
CFTR function in healthy and diseased cells. CFTR transports Cl^−^ and HCO_3_^−^ from inside the cell to outside the cell in epithelial cells in the lungs. This movement of anions forces Na^+^ and water to move in the same direction as Cl^−^ and HCO_3_^−^. This water is then used to hydrate the mucosal layer of the lungs. The lack of a functional CFTR protein causes the mucosal layer to dehydrate and shrink. Not shown in this figure is the movement of water, sodium, and bicarbonate [10]. Reprinted from “An Intriguing Involvement of Mitochondria in Cystic Fibrosis,” by M. Favia, L. de Bari, A. Bobba, and A. Atlante, 2019, *Journal of Clinical Medicine*, 8(11), p. 1890 (https://doi.org/10.3390/jcm8111890). CC BY-NC.

**Figure 2 biomimetics-09-00574-f002:**
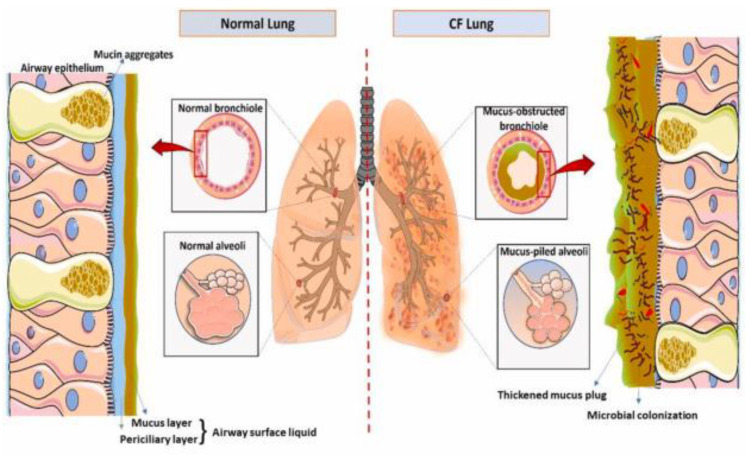
Normal lung vs. CF lung [23]. Reprinted from “Role of Innate Immunity and Systemic Inflammation in Cystic Fibrosis Disease Progression” by A.K. Purushothaman and E.J.R. Nelson, 2023, *Heliyon*, 9(7), p. 5 (https://doi.org/10.1016/j.heliyon.2023.e17553). CC BY-NC.

**Figure 3 biomimetics-09-00574-f003:**
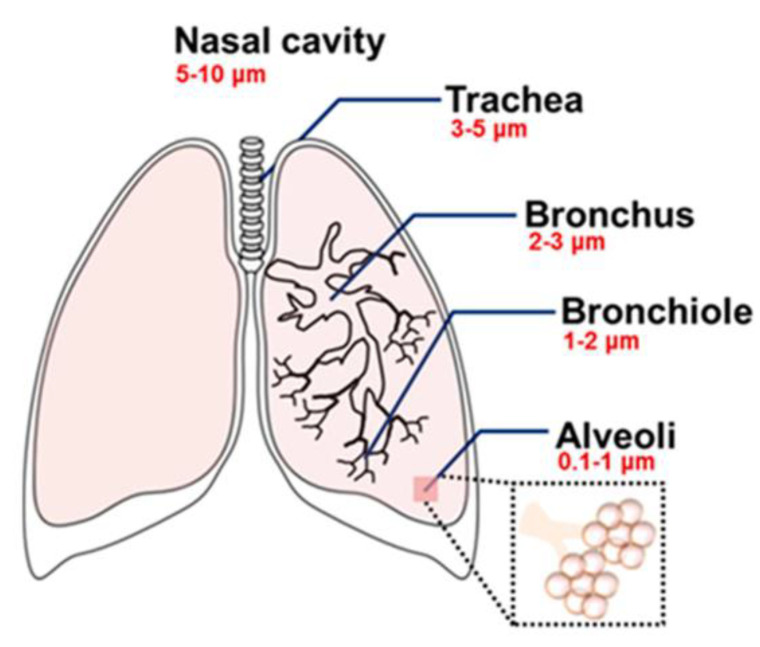
Deposition of particles in the lungs based on size. Inhaled particles deposit in the lungs primarily based on size. NPs 5–10 μm will only make it as far as the nasal cavity. NPs 3–5 μm will make it as far as the trachea, NPs 2–3 μm as far as the bronchus, and NPs 1–2 μm as far as the bronchiole. NPs 1–1000 nm will make it as far as the alveoli, though they will deposit everywhere in the lungs [29]. Reprinted from “Exposure to Inorganic Nanoparticles: Routes of Entry, Immune Response, Biodistribution and In Vitro/In Vivo Toxicity Evaluation,” by V. De Matteis, 2017, *Toxics*, 5(4), p. 29 (https://doi.org/10.3390/toxics5040029). CC BY-NC.

**Figure 4 biomimetics-09-00574-f004:**
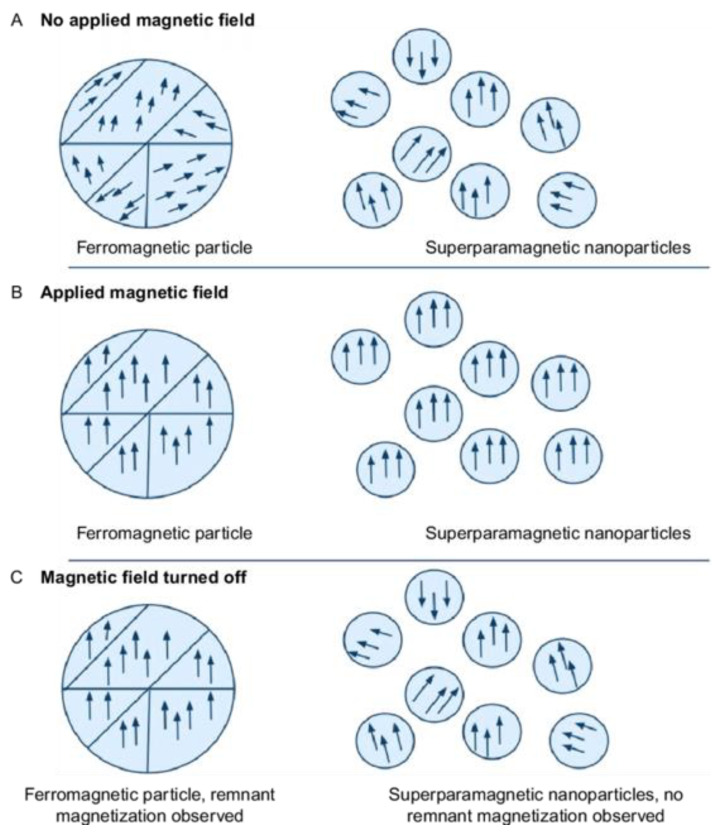
Ferromagnetic particles and SPIONs. Change in magnetic moments of ferromagnetic particles versus superparamagnetic particles. (**A**) In the absence of a magnetic field, ferromagnetic particles have magnetic moments in many directions, often aligned with their grain, while SPIONs have a single magnetic moment. (**B**) In the presence of a magnetic field, ferromagnetic particles and SPIONs align their magnetic moments to be parallel with the external magnetic field. (**C**) Upon removal of the external magnetic fields, ferromagnetic particles retain some of the magnetization as they dissipate energy and slowly return to the state present in A. SPIONs immediately release their magnetization and return back to the state present in A [48]. Reprinted from “Chapter 11—Magnetic Nanoparticles for Application in Biomedical Sensing”, by D. Alcantara and L. Josephson, 2012, *Frontiers of Nanoscience*, Vol. 4, p. 270 (https://doi.org/10.1016/B978-0-12-415769-9.00011-X). Copyright 2012 by Elsevier. Reprinted with permission.

**Figure 5 biomimetics-09-00574-f005:**
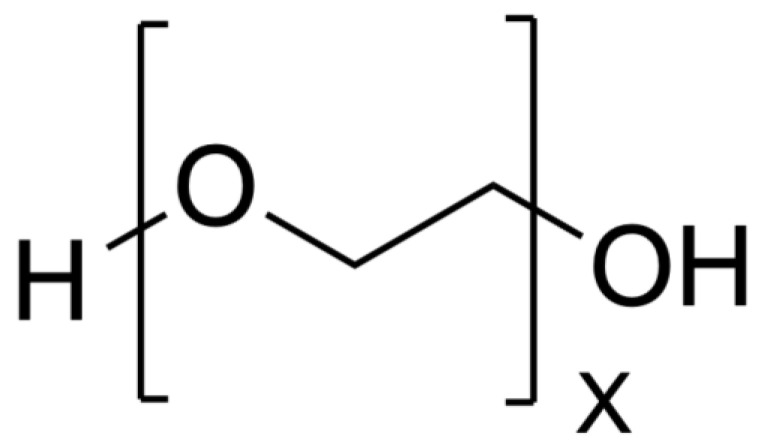
PEG structure [64].

**Figure 6 biomimetics-09-00574-f006:**
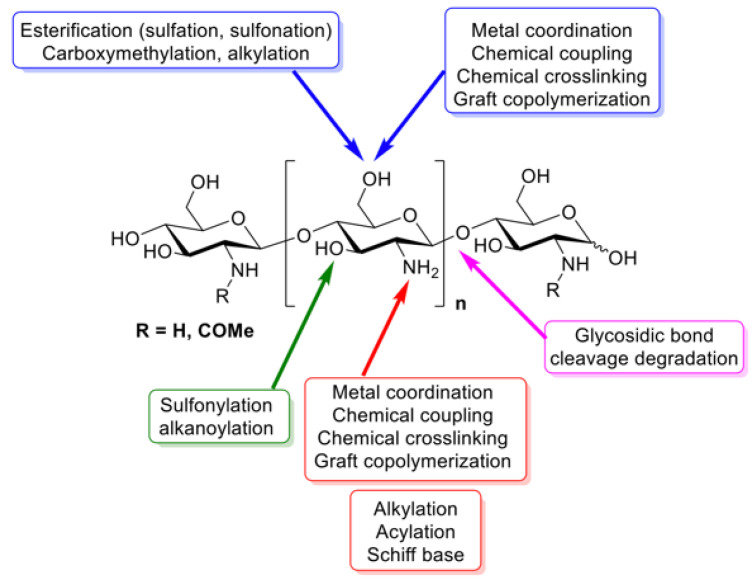
Groups in chitosan that can be functionalized [70]. Reprinted from “Chitosan: An Overview of Its Properties and Applications,” by I. Aranaz, A. Alcántara, M.C. Civera, C. Arias, B. Elorza, A.H. Caballero, and N. Acosta, 2021, *Polymers*, 13(19), p. 3256 (https://doi.org/10.3390/polym13193256). CC BY-NC.

**Figure 7 biomimetics-09-00574-f007:**
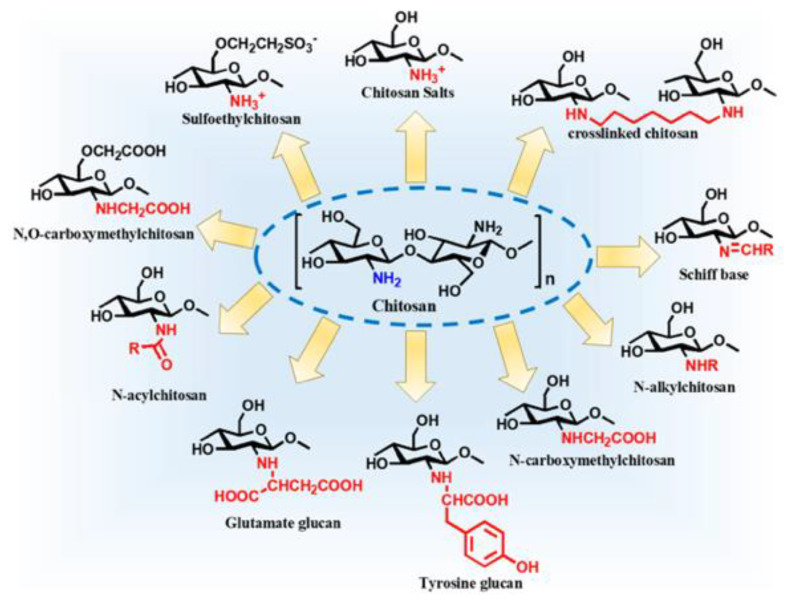
Chitosan derivatives [69]. Reprinted from “Chitosan Nanoparticles-Insight into Properties, Functionalization and Applications in Drug Delivery and Theranostics,” by J. Jhaveri, Z. Raichura, T. Khan, M. Momin, and A. Omri, 2021, *Molecules*, 26(2), p. 272 (https://doi.org/10.3390/molecules26020272). CC BY-NC.

**Figure 8 biomimetics-09-00574-f008:**
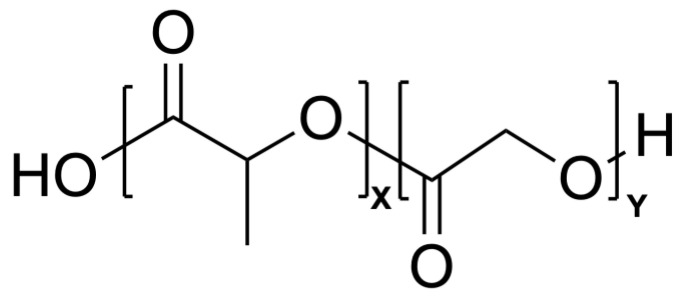
PLGA structure. Lactic acid monomers on left (x) and glycolic acid monomers on right (y) [64].

**Figure 9 biomimetics-09-00574-f009:**
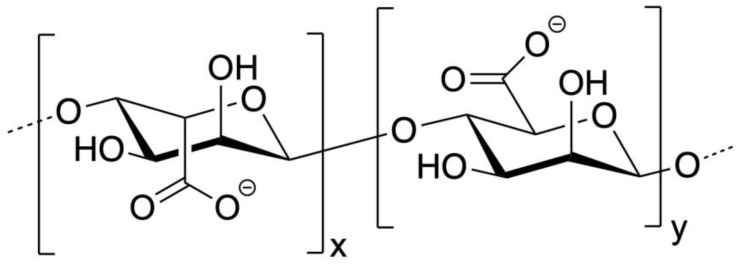
Alginate structure. Sodium ions coordinate around COO^−^ groups in sugars. G on left, M on right [64].

**Figure 10 biomimetics-09-00574-f010:**
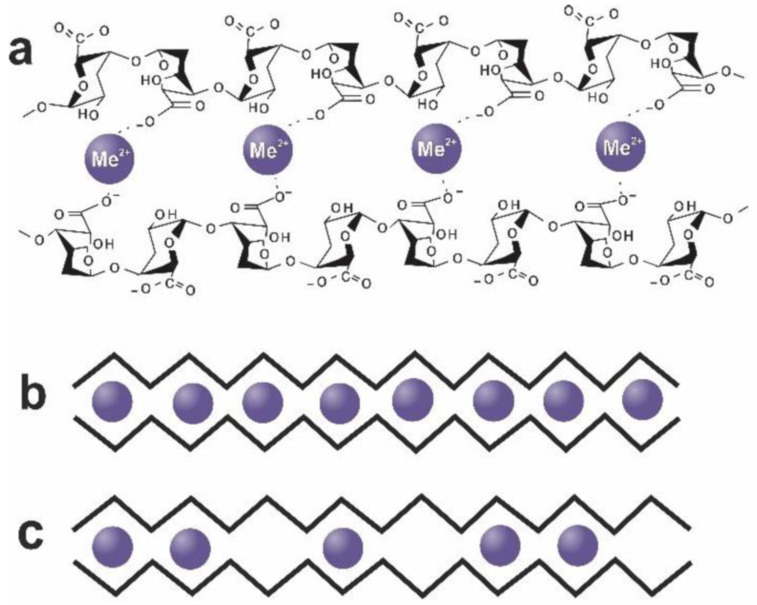
Alginate egg-box model (**a**) Ionic crosslinking as depicted on the molecular level. (**b**) Schematic view of crosslinking with full crosslinking across all alginate molecules. (**c**) Schematic view of crosslinking with partially crosslinked alginate molecules [87]. Adapted from “Ion-Induced Polysaccharide Gelation: Peculiarities of Alginate Egg-Box Association with Different Divalent Cations,” by A. Makarova, S. Derkach, T. Khair, M. Kazantseva, Y. Zuev, and O. Zueva, 2023, *Polymers*, 15(5), p. 1243 (https://doi.org/10.3390/polym15051243). CC BY-NC.

**Figure 11 biomimetics-09-00574-f011:**
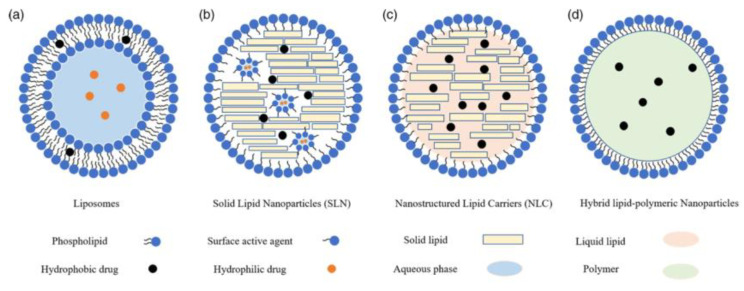
Types of lipid-based NPs. (**a**) Liposomes are formed by phospholipids and may encapsulate drugs in the hydrophilic core or hydrophobic section of the tails. (**b**) Solid Lipid Nanoparticles are micelles formed with a surface active agent (typically a phospholipid) along with a solid lipid. Hydrophobic drugs may be encapsulated in the solid lipid phase, or a smaller micelle may form inside the particle to encapsulate hydrophilic drugs. (**c**) Nanostructured lipid carries are micelles formed with a surface active agent (typically a phospholipid) along with both solid and liquid phase lipids. Hydrophobic drugs are encapsulated within the solid and liquid phase lipid. (**d**) Hybrid lipid-polymeric NPs are micelles formed by encapsulating a polymeric NP with phospholipid. Hydrophobic drugs are encapsulated by the polymer [102]. Reprinted from “Lipid Nanoparticles for Drug Delivery,” by L. Xu, X. Wang, Y. Liu, G. Yang, R. Falconer, and C. Zhao, 2021, *Advanced NanoBiomed Research*, 2(2) (https://doi.org/10.1002/anbr.202100109). CC BY-NC.

**Figure 12 biomimetics-09-00574-f012:**
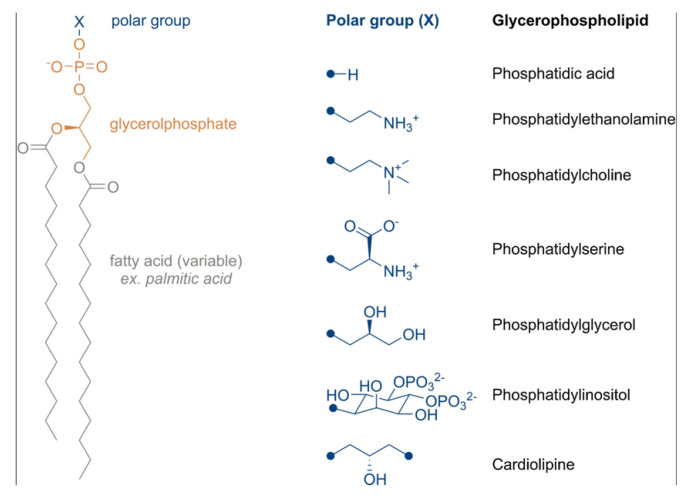
Phospholipid polar head group modifications [104]. Reprinted from “Liposome surface modification by phospholipid chemical reactions,” by P. de Lima, A. Butera, L. Cabeça, and R. Ribeiro-Viana, 2021, *Chemistry and Physics of Lipids*, 237 (https://doi.org/10.1016/j.chemphyslip.2021.105084). Copyright 2021 by Elsevier. Reprinted with permission.

**Figure 13 biomimetics-09-00574-f013:**
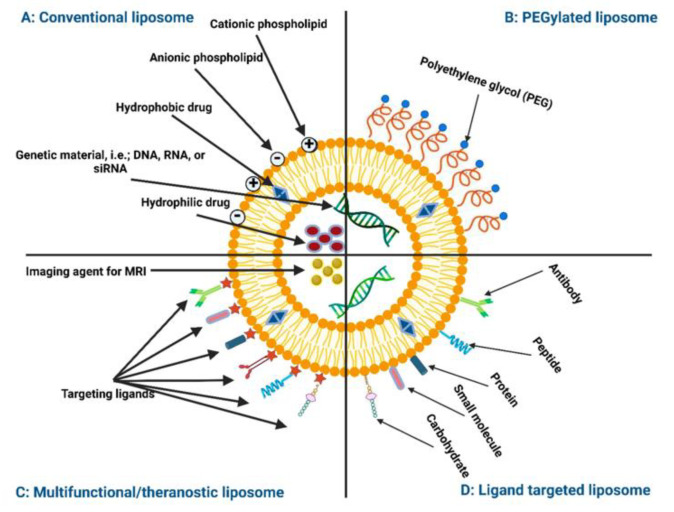
Lipid-NP functionalization [106]. Reprinted from “Liposomes or Extracellular Vesicles: A Comprehensive Comparison of Both Lipid Bilayer Vesicles for Pulmonary Drug Delivery,” by A. Al-Jipouri, S. Almurisi, K. Al-Japairai, L. Bakar, and A. Doolaanea, 2023, *Polymers*, 15(2), p. 318 (https://doi.org/10.3390/polym15020318). CC BY-NC.

**Table 1 biomimetics-09-00574-t001:** Metallic NPs.

Material	Conjugate	Model	Reference
Silver	N/A	*P. aeruginosa*	[52]
	Citrate-Capped	*P. aeruginosa*	[53]
	N/A	*P. aeruginosa*	[54]
Gold	N-Acetyl-Cysteine	*P. aeruginosa*	[55]
	Fucose or Galactose	*P. aeruginosa*	[56]
SPION	Tobramycin	*P. aeruginosa*	[57]
	Alginate and/or Tobramycin	*P. aeruginosa*	[58]
	Fluorescein or Complex Plasmid DNA	Mice	[59]

**Table 2 biomimetics-09-00574-t002:** Polymeric nanoparticles.

Material	Conjugate	Model	Reference
PEG	DNA NPs	Mice	[89]
PEG-PLGA	Bortezomib	Mice	[90]
PLGA	Curcumin	Mice	[91]
	Ciprofloxacin	*P. aeruginosa*	[92]
Chitosan	Alginate Lyase and Ciprofloxacin	Mice	[93]
	DNA Plasmid	Human Bronchial CF Cells	[94]
Alginate–Chitosan	Rifampicin and Ascorbic Acid	*S. aureus*	[95]
Alginate	Nitric Oxide-Releasing Conjugates	*P. aeruginosa*, *S. aureus*, and others	[96]
	Tobramycin and Dornase Alfa	*P. aeruginosa*	[97]

**Table 3 biomimetics-09-00574-t003:** Lipid nanoparticles.

Type of Lipid NP	Excipient	Model	Reference
Liposome	Chemically Modified mRNA	Bronchial Epithelial Cells, Mouse	[108]
	mRNA	CF-Based Cells, Mice	[109]
Micelle	PEG and siRNA	Artificial CF Mucus	[110]
NLC	Lumacaftor and Ivacaftor	Mice	[111]
SLNPs and NLCs	Sodium Colistimethate	Mice, *P. aeruginosa*	[112]
SLNP	Tween-80 or PVA	Artificial CF Mucus, CF Sputum, Mucus-Secreting Cell Line	[113]

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
