# Peer review of "Nanoparticles as Drug Delivery Vehicles for People with Cystic Fibrosis"

_biomimetics, 2024, doi:10.3390/biomimetics9090574_

Round 1
Reviewer 1 Report
Comments and Suggestions for Authors
This manuscript provides a comprehensive review of the use of nanoparticles as drug delivery vehicles for people with cystic fibrosis. The paper is well-structured, addressing the necessary background on cystic fibrosis, the challenges of its treatment, and the applications of different types of nanoparticles. The following points need to be addressed:
1. FDA-Approved Nanoparticles section needs to be expanded. Expanding this section to give more detail on the clinical trials, efficacy, and safety of these nanoparticles would add depth to the review. Information can be added in the form of a table or in text.
2. Some lines are repeatedly used in the text; for eg.: discussion of CFTR function and the impact of CF on lung function is repeated in different sections.
3. The whole text should be checked for possible grammatical and typographical errors.
4. All the references are not uniformly written, for eg. like reference no 23; In few references, the title is in capital whereas others are in small. It is recommended to prepare reference section in uniform pattern as per journal guidelines.
Comments on the Quality of English LanguageThe paper is well written but needs revision as recommended.
Author Response
Dear Editors and Reviewers,
We are pleased to submit the revised version of Biomimetics (ID# 3156484): “Nanoparticles as Drug Delivery Vehicles for People with Cystic Fibrosis.” We appreciate the reviewers' valuable feedback and have thoroughly addressed each of their concerns as detailed below. All revisions to the manuscript are indicated as tracked changes. Additionally, we have included a clean version of the manuscript without the tracked changes. We enhanced both the text and figures, and added additional information to enrich the paper; these modifications have notably enhanced the paper and improved its clarity.
Reviewer 1:
- FDA-Approved Nanoparticles section needs to be expanded. Expanding this section to give more detail on the clinical trials, efficacy, and safety of these nanoparticles would add depth to the review. Information can be added in the form of a table or in text.
Response: We agree with the reviewer that the addition of information surrounding the clinical trials is valuable. We have added text on the (three) phase III clinical trials for TOBI PODHALER, and the (one) phase III clinical trial for ARIKAYCE (lines: 803-881).
- Some lines are repeatedly used in the text; for eg.: discussion of CFTR function and the impact of CF on lung function is repeated in different sections.
Response: Much of the manuscript has been edited to ideally cut down on this; however, repetition of certain methods is useful to help the reader understand the direct connections of the different NPs to CF. Further, many of the images may require something to be restated so the mechanism depicted is clear. We hope these edits are sufficient to reduce repetition while providing clarity.
- The whole text should be checked for possible grammatical and typographical errors.
Response: The text has been edited to correct grammatical and typographical errors.
- All the references are not uniformly written, for eg. like reference no 23; In few references, the title is in capital whereas others are in small. It is recommended to prepare reference section in uniform pattern as per journal guidelines.
Response: The references have been altered in reference manager to align with MDPI requirements. To the best of our knowledge, all references are now in line with the journal guidelines and all paper titles are no longer in capital letters.
Reviewer 2 Report
Comments and Suggestions for Authors
This paper is very clear and focus to the CF topic. The discussion is easy to understand. Reviews on different materials used in nanoparticles formulations is smooth and brilliant. Moreover, the pictures are clear and sharp.
Author Response
Dear Editors and Reviewers,
We are pleased to submit the revised version of Biomimetics (ID# 3156484): “Nanoparticles as Drug Delivery Vehicles for People with Cystic Fibrosis.” We appreciate the reviewers' valuable feedback and have thoroughly addressed each of their concerns as detailed below. All revisions to the manuscript are indicated as tracked changes. Additionally, we have included a clean version of the manuscript without the tracked changes. We enhanced both the text and figures, and added additional information to enrich the paper; these modifications have notably enhanced the paper and improved its clarity.
Reviewer 2:
Comments and Suggestions for Authors: This paper is very clear and focus to the CF topic. The discussion is easy to understand. Reviews on different materials used in nanoparticles formulations is smooth and brilliant. Moreover, the pictures are clear and sharp.
Response: We greatly appreciate the reviewer’s comments and support of our paper.
Reviewer 3 Report
Comments and Suggestions for Authors
The manuscript is titled "Nanoparticles as Drug Delivery Vehicles for People with Cystic Fibrosis ". This work aims to present an overview of cystic fibrosis (CF) and its effects on the lungs, as well as the obstacles to treating the condition with drug-delivery vehicles made of nanoparticles. Finally, it reviews the three material classes that are frequently used in nanoparticle formulations: lipids, polymers, and metals. However, it is crucial to offer further clarification on certain issues Therefore, it is recommended that this manuscript be published with only minor revisions.
In line 719 Figure 12 is not mentioned in the text.
In line 169, the Complete name of NP should be specified.
In line 489, the Complete name of IUPAC should be specified.
Author Response
Dear Editors and Reviewers,
We are pleased to submit the revised version of Biomimetics (ID# 3156484): “Nanoparticles as Drug Delivery Vehicles for People with Cystic Fibrosis.” We appreciate the reviewers' valuable feedback and have thoroughly addressed each of their concerns as detailed below. All revisions to the manuscript are indicated as tracked changes. Additionally, we have included a clean version of the manuscript without the tracked changes. We enhanced both the text and figures, and added additional information to enrich the paper; these modifications have notably enhanced the paper and improved its clarity
Reviewer 3:
In line 719 Figure 12 is not mentioned in the text.
Response: We have added a reference to Figure 12 (now Figure 13) on line 755 and moved the figure below this paragraph.
In line 169, the Complete name of NP should be specified.
Response: We have removed all instances of the word “nanoparticles” except the first occurrence both within the abstract and the main body paragraphs, or when it is introduced as part of a longer abbreviation. This has been done to maintain consistency throughout the paper as suggested.
In line 489, the Complete name of IUPAC should be specified.
Response: The specified abbreviation “IUPAC” was edited to include the complete name of “International Union of Pure and Applied Chemistry” (line 506).
Reviewer 4 Report
Comments and Suggestions for Authors
Dear Authors,
Thank you very much for submitting the review manuscript entitled "Nanoparticles as Drug Delivery Vehicles for People with Cystic Fibrosis".
Please, find my suggestions and comments below.
1. P. 3, line 110, section 2
I kindly recommend adding the schematic picture, which demonstrates normal (healthy) lung and pathologic lung (the pulmonary mucosal layer should be highlighted).
2. P. 4, line 179, subsection 3.1. Pulmonary Deposition and P. 5, line 220, subsection 3.2. Pulmonary Deposition
Subsection 3.2 should be renamed (e.g. Mucus Penetration). Please, check.
3. P. 7, section 4 Metallic Nanoparticles
I kindly recommend the Authors adding several recent examples of metallic nanoparticles used for drug delivery for patients with cystic fibrosis.
4. P. 9, section 5, lines 384 - 395
This paragraph contains known information and should be excluded.
At the same time, please, use the correct classification of polymers: natural, synthetic, and semisynthetic polymers (3 types). This classification is more preferable. Please, rewrite the second paragraph.
5. P. 11, Figure 5 and 6
Figures should be scaled up.
6. P. 14, Figure 9
Figure should be scaled up.
7. The Authors give high attention to the polymers, and their properties. At the same time, this review aims to collect data related to nanoparticles. For this reason, I kindly recommend the Authors to shift focus to the recent examples of nanoparticles which were elaborated earlier. Moreover, the summary table with examples discussed should be added after section 4, 5, and 6.
8. P. 21, line 871, References
Please, use MDPI style for the references.
9. Please, highlight the novelty and advantages of this review in comparison with the existent.
In summary, in current form this manuscript is not recommended for publication.
Author Response
Dear Editors and Reviewers,
We are pleased to submit the revised version of Biomimetics (ID# 3156484): “Nanoparticles as Drug Delivery Vehicles for People with Cystic Fibrosis.” We appreciate the reviewers' valuable feedback and have thoroughly addressed each of their concerns as detailed below. All revisions to the manuscript are indicated as tracked changes. Additionally, we have included a clean version of the manuscript without the tracked changes. We enhanced both the text and figures, and added additional information to enrich the paper; these modifications have notably enhanced the paper and improved its clarity.
Reviewer 4:
- 3, line 110, section 2
I kindly recommend adding the schematic picture, which demonstrates normal (healthy) lung and pathologic lung (the pulmonary mucosal layer should be highlighted).
Response: As recommended by the reviewer, a new Figure 2 has been inserted to provide a graphical illustration of the differences between a normal (healthy) and CF (diseased) lung.
- 4, line 179, subsection 3.1. Pulmonary Deposition and P. 5, line 220, subsection 3.2. Pulmonary Deposition
Subsection 3.2 should be renamed (e.g. Mucus Penetration). Please, check.
Response: We appreciate the reviewer catching this and Subsection 3.2 has been renamed appropriately.
- 7, section 4 Metallic Nanoparticles
I kindly recommend the Authors adding several recent examples of metallic nanoparticles used for drug delivery for patients with cystic fibrosis.
Response: A table with this information has been added to the end of section 4 (Table 1. Metallic NPs).
- 9, section 5, lines 384 - 395
This paragraph contains known information and should be excluded.
At the same time, please, use the correct classification of polymers: natural, synthetic, and semisynthetic polymers (3 types). This classification is more preferable. Please, rewrite the second paragraph.
Response: A large portion of paragraph one has been removed, except for the first few sentences defining what polymers are as this is important background information and provides a transition from the previous section. The classification of polymers has been changed to natural, synthetic, and semisynthetic with definitions for each to be in line with modern classifications.
- 11, Figure 5 and 6
Figures should be scaled up.
Response: We have scaled up the figures, as suggested, due to their quality.
- 14, Figure 9
Figure should be scaled up.
Response: We have scaled up the figures, as suggested, due to their quality.
- The Authors give high attention to the polymers, and their properties. At the same time, this review aims to collect data related to nanoparticles. For this reason, I kindly recommend the Authors to shift focus to the recent examples of nanoparticles which were elaborated earlier. Moreover, the summary table with examples discussed should be added after section 4, 5, and 6.
Response: While we appreciate the reviewer’s comment regarding manuscript emphasis, this review paper focuses less on specific instances of the materials discussed and, instead, more on the theory behind material function, why or how that material might be used, and how it may be preferable to other materials for CF. Other manuscripts have focused on recent examples of nanoparticles more thoroughly and, to maintain both novelty and usefulness of a review in the CF space, we wish to maintain our focus. Related to the suggestion on tables, a summary table presenting specific examples of discussed NPs has been added to the end of sections 4, 5, and 6 (Tables 1-3, respectively).
- 21, line 871, References
Please, use MDPI style for the references.
Response: The references have been altered in reference manager to align with MDPI requirements. To the best of our knowledge, all references are now in line with the journal guidelines and all paper titles are no longer in all capital letters.
- Please, highlight the novelty and advantages of this review in comparison with the existent.
Response: The abstract has been updated to highlight that this review paper focuses less on specific examples/instances of the materials discussed, and instead on the theory behind material function, why or how that material might be used, and how it may be preferable to other materials. (Lines 17-19)
Round 2
Reviewer 4 Report
Comments and Suggestions for Authors
Dear Authors,
I am very thankful to you for the revised manuscript.
The study was fully corrected in accordance with my suggestions.
I recommend accepting this study as is.